# Lipid−lncRNA Crossroads: An Overview of Interactions Between Lipids and lncRNA

**DOI:** 10.3390/cells14151193

**Published:** 2025-08-02

**Authors:** Andrea Bayona-Hernandez, Ana Miladinović, Ludovica Antiga, Pavel Hozak, Martin Sztacho, Enrique Castano

**Affiliations:** 1Unidad de Biologia Integrativa, Centro de Investigación Científica de Yucatán, A.C., Mérida 97205, Yucatan, Mexico; andrea.bayona.her@gmail.com; 2Department of Biology of the Cell Nucleus, Institute of Molecular Genetics of the Czech Academy of Sciences, 142 20 Prague, Czech Republic; ana.miladinovic@img.cas.cz (A.M.); pavel.hozak@img.cas.cz (P.H.); 3Laboratory of Cancer Cell Architecture, Institute of Biochemistry and Experimental Oncology, First Faculty of Medicine, Charles University, 128 00 Prague, Czech Republic; martin.sztacho@lf1.cuni.cz

**Keywords:** lncRNA, phospholipids, phase separation

## Abstract

Long non-coding RNAs (lncRNAs) interact with a variety of biomolecules, including DNA, mRNAs, microRNA, and proteins, to regulate various cellular processes. Recently, their interactions with lipids have gained increasing attention as an emerging research area. Both lipids and lncRNAs play central roles in cellular regulation, and growing evidence reveals a complex interplay between these molecules. These interactions contribute to key biological functions, such as cancer progression, lipid droplet transport, autophagy, liquid−liquid phase separation, and the formation of organelles without membranes. Understanding the lipid−lncRNA interface opens new avenues for unraveling cellular regulation and disease mechanisms, holding great potential not only for elucidating the fundamental aspects of cellular biology but also for identifying innovative therapeutic targets for metabolic disorders and cancer. This review highlights the biological relevance of lipid–lncRNA interactions by exploring their roles in cellular organization, regulation, and diseases, including metabolic and cancer-related disorders.

## 1. Introduction

Lipids and long non-coding RNAs (lncRNAs) play pivotal roles in the intricate dynamics of cellular processes. Growing evidence suggests multifaceted interactions between these two distinct biomolecular components. This review aims to provide a comprehensive overview of the current understanding of the interactions and functions of lipids and lncRNAs, highlighting the significance of this relationship in various biological contexts.

LncRNAs are RNA molecules above 200 nucleotides that have gathered increasing attention in recent years, as they have been found to play crucial regulatory roles in a wide range of cellular processes; they are themselves the functional units, in which their subcellular localization is critical to their function [1]. Notably, the functional annotation of lncRNAs has revealed their involvement in regulating distinct biological pathways, transcriptional machinery, alternative promoter activity, genome architecture, and cellular organization [2,3]. LncRNAs are involved in telomere maintenance, epigenetic regulation, alternative splicing, and scaffolding of ribonucleoprotein complexes [4]. For example, the lncRNA telomerase RNA component (TERC) plays an essential role in telomerase activity by providing an RNA template for telomere elongation [5]. Other functions of lncRNAs include involvement in lipid metabolism and homeostasis [6,7,8].

Lipids are a diverse class of biomolecules that play pivotal roles in cellular structure, signaling, and metabolism [9,10]. In particular, phospholipids influence the localization and stability of lncRNAs, alter nuclear structure, and affect numerous nuclear functions. Specific lipid−lncRNA interactions can target lncRNAs to specific subcellular compartments. The discovery and functional study of lipid-associated lncRNAs could help broaden our understanding of the regulatory mechanisms in diseases such as cancer, as many of these lncRNAs are known to be involved, some in mechanisms that are not yet understood [11,12,13]. Cell membrane-associated lncRNAs can act as mediators of signaling pathways and serve as critical components of signal transduction cascades and gene networks that modulate cancer development and progression [14].

To appreciate the interplay between lncRNAs and lipids, we first summarize the characterized regulatory effects of lncRNAs on lipid metabolism, even in the absence of direct binding. This overview sets the stage for a more detailed discussion of physical lncRNA and lipid interactions, which form the core focus of this review. LncRNAs are associated with lipids in several ways, including involvement in lipid metabolism and homeostasis. They also participate in direct interactions involved in biological processes such as lipid droplet transport and cancer-related pathways. In the following sections, we discuss some of these lncRNAs.

## 2. Functional Roles of lncRNAs in Lipid Metabolism

Long non-coding RNAs (lncRNAs) are key regulators of cellular development and gene expression, including lipid metabolism. Numerous lncRNAs influence cholesterol homeostasis through diverse regulatory mechanisms and have been implicated in lipid-related diseases [7]. This section highlights the findings on lncRNAs involved in lipid metabolism. The lncRNAs discussed below were identified from a dataset obtained by sequencing RNA with affinity for phospholipids PIP2 and PI4P, available under BioProject number PRJNA937906 [15]. Although their direct physical interactions with phospholipids remain to be fully characterized, they have been described to play roles in several molecular processes, even those related to lipid metabolism. Recent studies have highlighted the emerging roles of long non-coding RNAs in gene regulation and cancer biology, as well as in the modulation of lipid metabolism and associated cardiovascular diseases. HOXC-AS1, HOTAIR, and MALAT1 are lncRNAs that have gained attention due to their multifaceted involvement in metabolic and pathological processes.

The lncRNA HOXC cluster antisense RNA 1 (HOXC-AS1) is involved in several types of cancer, such as gastric cancer [16] and nasopharyngeal carcinoma (NPC) [17]. Furthermore, HOXC-AS1 has been implicated in cardiovascular diseases, particularly atherosclerosis [18,19]. A microarray assay of carotid atherosclerosis and normal renal arterial intima tissue showed that HOXC-AS1 was downregulated. An assay in THP-1 macrophages to enhance lncRNA levels showed that HOXC6 levels were increased at the mRNA and protein levels (Figure 1a). Ox-LDL treatment of THP-1 macrophages promoted cholesterol accumulation and increased the levels of total cholesterol, free cholesterol, and cholesterol esters. Interestingly, overexpression of lncRNA HOXC-AS1 partially reduced cholesterol accumulation, indicating a role for this lncRNA in cholesterol homeostasis. These findings suggest that HOXC-AS1 may be a potential therapeutic target for atherosclerosis [18].

Similarly, the lncRNA HOX transcript antisense RNA (HOTAIR), which is characterized by its role in chromatin remodeling and cancer progression, has also been linked to lipid metabolism. HOTAIR is an lncRNA involved in several cellular processes, including regulation of gene expression, epigenetic modifications, and cancer progression through mechanisms that alter chromatin remodeling, transcriptional regulation, and post-transcriptional processing [20,21,22]. Recent studies have shown that HOTAIR may also play a role in lipid metabolism, which is expressed in gluteal adipocytes, and overexpression of HOTAIR was observed to promote the differentiation of abdominal preadipocytes into adipocytes [23]. HOTAIR is induced in the early stages of adipogenesis and also plays a role in regulating translation and cytoskeletal reorganization during early adipogenesis in abdominal adipose stem cells (ASCs), acting as a determinant of lipid storage capacity in adipocytes [22]. This lncRNA also plays a role in atherosclerosis by downregulating miR-19a-3p (Figure 1b) through HOTAIR overexpression and observing a negative correlation in the plasma of patients with atherosclerosis, and reducing the foam cell formation and inflammatory reaction [20].

Another lncRNA, MALAT1, has also emerged as a regulator of lipid metabolism. Studies have described that MALAT1 is upregulated in hepatic steatosis and insulin resistance, and its levels increase in hepatocytes exposed to palmitate, which produces lipid accumulation, as well as in obese mouse models, ob/ob mice. Its overexpression can exacerbate these metabolic disorders by increasing sterol regulatory element-binding protein (SREBP)-1c, a transcriptional factor of genes related to hepatic lipogenesis. MALAT1 overexpression not only increases the nuclear accumulation of SREBP-1c but also stabilizes it by preventing its ubiquitin-mediated degradation (Figure 1c) [24].

Although the lncRNA mechanisms above do not directly discuss lipid binding, their participation in lipid metabolism through diverse mechanisms and their role in several metabolic diseases highlight the functional importance of the lncRNA–lipid relationship. This regulatory influence, even in the absence of physical binding, shows the biological significance of the lncRNA–lipid axis and provides important context for understanding the direct molecular interactions explored in the following sections.

## 3. Interaction Platforms for RNA−Lipid-Protein in Liquid−Liquid Interactions

Lipids are critical biomolecules that play diverse roles in cellular processes, ranging from being structural components of cell membranes to signaling molecules [10,25]. They are also involved in various cellular and organismal processes, such as energy metabolism, signal transduction, modulation of membrane fluidity, and regulation of transcriptional activities, and play a role in diseases, such as cancer, among other functions [25,26,27,28].

Although phosphoinositides constitute only about 5% of total phospholipids and are present in relatively low concentrations compared to other cellular lipids, they serve as critical signaling molecules. These lipids regulate a number of cellular processes, including cell division, vesicle trafficking, transcription, cell differentiation, viral transduction, proliferation, and the formation of phase-separated condensates, among other essential functions [29,30,31,32,33].

PI4P has been detected in the nucleus, where it localizes to nuclear speckles and forms distinct foci in the nucleoplasm [34]. Also, Phosphatidic acid and phosphatidylinositol 4-phosphate (PI4P) are crucial for interactions that mediate lipid droplet (LD) transfer [13].

While most roles have been explored for cytoplasmic phospholipids, they are also present in the nucleus and its various membrane-less compartments, the most abundant being phosphatidylinositol 4,5-bisphosphate, phosphatidylinositol 3,4-bisphosphate, and PI4P in nuclear speckles, nucleoplasm, and nucleoli, and nuclear lipid islets (NLIs) [35,36,37]. Phosphatidylinositol 4,5-bisphosphate, hereafter referred to as PIP2, is a key lipid messenger that modulates transcription by interacting with various transcription factors and chromatin remodeling complexes. PIP2 directly binds to and regulates the activity of specific transcription factors and recruits chromatin-modifying enzymes to influence gene expression programs. The spatial and temporal dynamics of PIP2 signaling within the nucleus play critical roles in fine-tuning transcriptional activity and coordinating cellular processes. These structures may serve as platforms for the assembly of the transcriptional machinery, facilitating the formation of a transcriptionally competent nuclear architecture [33,36,38].

Phosphatidylinositol-3,4,5-trisphosphate (PIP3) functions as a lipid second messenger and is indispensable for signal transduction at the plasma membrane. It also facilitates the recruitment and subsequent activation of proteins containing pleckstrin homology domains, such as AKT, various exchange factors, and kinases [39]. Nuclear PIP3 is a pivotal regulator of cellular differentiation and anti-apoptotic signaling. Furthermore, PIP_3_ interacts with non-coding RNAs and various proteins to modulate diverse signaling pathways, and impaired binding within these interactions has been correlated with tumorigenesis [34].

Recent studies have demonstrated that membrane lipid domains can serve as platforms for the recruitment and interaction of lncRNAs and their associated proteins, facilitating the formation of membrane-less organelles and liquid−liquid phase separation (LLPS) events [40,41]. The formation of cellular compartments, particularly in the nucleus, is often driven by the interactions between molecules such as proteins, lipids, and nucleic acids. These interactions can lead to phase-separation processes and the formation of subcellular bodies that often have specific functions within the cell [42]. Phosphoinositides, such as PIP2, provide a platform for the recruitment of interacting proteins, allowing them to increase their local concentration. The formation of liquid−liquid phase-separated compartments in the nucleus can also be driven by the presence of these phosphoinositides [36,39]. Studies have shown that the presence of PIP2 in the nucleus, and its spatial colocalization with other molecules such as fibrillarin, depend on the involvement of RNA and are also linked to the activity of RNA polymerase I/II for transcription [35,36,37,43].

Cellular condensates form due to factors sharing chemical and physical properties, such as multivalency, highly charged regions, and enrichment in intrinsically disordered sequences. Intrinsically disordered proteins (IDPs), a key group of molecules involved in liquid–liquid phase separation, are composed of amino acids that do not fold into a defined three-dimensional structure. [44]. Phase separation allows cells to efficiently organize multiple processes with minimal energy use by enabling the formation of these subcellular compartments. Chemical modifications of proteins, RNA, and lipids help to concentrate enzymatic reactions in specific regions by altering the local molecular environment [45].

Condensates are functional compartments formed not only by RNA and proteins, but also by lipids, which help create the chemical microenvironment and open the possibility of facilitating phospholipid biology and signaling. It has been suggested that phospholipids can modulate condensate characteristics, impacting both size and morphology. A study of the formation of condensates was analyzed using specific proteins: full-length SARS-CoV-2 nucleocapsid, MED1, and HNRNPA1, which contain low-complexity domains that exhibit the capacity to engage in weak, non-specific interactions with RNA molecules [46]. The analysis was made using a method for inducing phase separation compatible with liquid chromatography–mass spectrometry (LC-MS) that allows the identification of the different metabolites that partition in the different condensates, showing the partition of similar metabolites despite the different aminoacidic composition, and it was also found that diverse phospholipids can partition into these microenvironments [46].

Analysis of the nuclear RNA-dependent PIP2-associated (RDPA) proteome identified approximately 160 proteins containing charged K/R motifs within their intrinsically disordered regions. Among these, Bromodomain-containing protein 4 (BRD4) plays a crucial role in gene expression and contributes to the compartmentalization and concentration of the transcriptional machinery. Treatment with RNase III reduced PIP2 presence in nuclear speckles, while assays modulating the enzymes that regulate PIP2 levels demonstrated that elevated PIP2 levels increased the number of BRD4 nuclear foci [33].

## 4. Lipid−lncRNA Interactions

Studies of RNA and lipid interactions have increased in recent times, and there have been studies describing these interactions and the biological functions that reside on them, for example, the lncRNA LINK-A that interacts with PIP3 that leads to AKT inhibitors resistance [47], or the cancer-related lncRNA−lipid-droplet transporter (LIPTER) that has a role in modulating on the lipid droplets and its interaction with PIP4 and PA [13].

To explore the regulation of RNA activity on lipid membranes, it has been reported that RNA−lipid interactions are dependent on nucleotide composition, base pairing, and length. By employing phosphatidylcholine lipids, the necessary properties for RNA binding to membranes were analyzed, emphasizing that RNA sequences abundant in guanines demonstrated greater binding rates, particularly to dipalmitoylphosphatidylcholine (DPPC) gel membranes. Base pairing is regarded as playing a crucial role in membrane contacts and serves as a key factor in the creation of secondary structures. In experiments involving dsRNA with repeated complementary sequences and ssRNA, dsRNA exhibited membrane-binding capacity; however, it interacted with fewer lipids than ssRNA. This indicates that varying binding models can influence the selectivity of RNA−lipid interactions [48]. An analysis to select RNAs that could interact with lipid membranes and identify the molecular characteristics that allow this interaction revealed that RNAs form multimeric complexes via complementary loop interactions, enhancing their membrane-binding capabilities. Oligomerization increases RNA affinity for phospholipid membranes [49].

Extracellular vesicles are also closely related to RNA−lipid interactions. These are lipid-bilayer-delimited particles released by cells that are known to transport bioactive molecules, such as proteins, DNA, lipids, and RNA, including lncRNA and circular RNA (circRNA). They facilitate intercellular communication and play crucial roles in both physiological and pathological processes. Specific lncRNAs can be selectively packaged into EVs, where they can modulate gene expression in recipient cells at multiple levels, thereby influencing processes such as cancer progression [50]. RNA structural motifs have been found to modulate interactions with the plasma membrane, suggesting the importance of these motifs for the loading of lncRNAs into extracellular vesicles and RNA-based lipid biosensors. Analysis of Y RNA has revealed the presence of the UCCCU RAFT motif, which promotes lipid raft association, and the GGAG EXO motif, which facilitates miRNA sorting into exosomes. Furthermore, the presence of multiple small apical loops within the RNA structure was associated with an enhanced affinity for the RAFT liposomes. Also, it was found that viral RNA fragments with a long double helix at the apical loop increase the affinity of viral RNA to lipid rafts [51].

Using a different approach to analyze membrane-associated RNAs, the APEX-seq method was used to investigate RNA interactions with the plasma membrane. This analysis identified 75 RNAs associated with the plasma membrane in mammalian cells. PMART72 is particularly noteworthy because of its ability to bind to multiple lipids. Further investigation using a fat blot assay revealed that PMART72 exhibited the highest affinity for sphingomyelin (SM), followed by cholesterol and PIP3 [52].

Current research on the structure of snRNAs that interact with lipids has revealed some possible mechanisms of interaction between RNA and lipids. For example, snR191 is composed of a multi-branch loop, a unique tetraloop motif (UUGG), and an H/ACA motif. Interaction with PIP2 and snR191 was previously shown [15]. The binding stability of snR191-PIP2 is mediated by the formation of six hydrogen bonds across four nucleotides (nt) of snR191 located in the region comprising the fifty nt sequence. The snR191-PIP2 interaction occurs through two hydrogen bonds between residues A31 and the oxygen-11 atom of the 2′-phosphate and A45 and the oxygen-4 atom of the 1′-phosphate of PIP2, whereas A43 and U44 form four H-bonds with the oxygen-16 and oxygen-17 atoms of PIP2, respectively. Such interactions may vary depending on the RNA structure and lipids [53].

Few studies have described phospholipid-RNA interactions. The first to be described is the long intergenic non-coding RNA for kinase activation (LINK-A), found in the lipidic fraction of triple-negative breast cancer (TNBC) patient tissues, which is characterized by an interaction with PIP3 and phosphatidylcholine and a weak interaction with PIP2. This was first validated using lipid strips, and the sequence was mapped to identify the sites responsible for phospholipid interactions. Deletion mutants revealed that nt 1081–1140 and 241–300 are the sites of PIP3 and PC interactions, respectively. By analyzing the sequence using an oligonucleotide corresponding to the 1100–1117 of the LINK-A sequence, 5′-CAGGGUAGACUCGCUCUG-3′ corresponding to the loop in the PIP3 binding region and nucleotide mutations was discovered LINK-A and PIP3 interaction using techniques as Alpha assays and RNA–PIPs overlay assay, it was found that in the central loop region, the nt A (1108) is responsible for AKT interaction while the C (1109) and C (1111) are important for the PIP3 binding [47].

Another lncRNA associated with breast cancer is the small nucleolar RNA host gene 9 (SNHG9), which correlates with breast cancer progression and other types of cancer and was found to interact with PA (phosphatidic acid) and Large Tumor Suppressor Kinase 1 (LATS1) [54,55,56]. Analysis of the sequence revealed the regions responsible for the interaction using in vitro transcribed SNHG9 and mutant deletions along the RNA sequence using a lipid−RNA dot blot assay. The loop of 204–231 nt, sequence detailed in Table 1, was found to be responsible for PA binding, and the two loops of 7–25 and 30–43 nt were found to be responsible for the LATS1 interaction site. These interactions were shown to promote the liquid−liquid phase separation of LATS1 [55].

In addition, LIPTER is essential for lipid droplet (LD) transport in cardiomyocytes and has been described to selectively interact with phospholipids on the surface of LD and with the MYH10 protein. The direct interaction of LIPTER with PA and PI4P was determined using an RNA−lipid overlay assay. This interaction with specific phospholipids suggests a role for LIPTER in the regulation of lipid droplet dynamics and transport within cardiomyocytes [13].

In the search for an aptamer with specific lipid binding, the Systematic Evolution of Ligands by Exponential Enrichment (SELEX) screening identified an aptamer RNA sequence specific for PI3P binding, a phosphoinositide that plays an important role in autophagy. This aptamer bound to PI3P in a pH and Mg^2+^ concentration-dependent manner without requiring Ca^2+^. The 40-nt aptamer was compared with the LINK-A PIP3 binding motif, revealing similarities in the sequence. Further investigation using mutation analysis and PIP strip assays revealed the importance of certain bases in the central region of the aptamer for its interaction with PI3P. Screening with substitution mutants showed a major loss of interaction in mutant II at positions 13 and 14 of the nt sequence [57].

The LIPRNAseq method was developed to understand and identify a wider range of interactions between lipids and RNA. This approach has been used to analyze RNA affinity to PIP2 across various organisms, including humans, plants, and yeast, revealing differentially enriched RNA populations in each case [15].

The lncRNA HANR interacts with PIP2 primarily within perinucleolar compartments (PNCs). These subnuclear structures, located at the nucleolar periphery, are enriched with PIP2. In addition, motif enrichment analysis of PIP2-bound RNAs identified two synthetic AU-rich RNA sequences, AU1 and AU2, obtained using previous data obtained by LIPRNAseq and MEME Suite for motif enrichment analysis. Both sequences exhibited specific interactions with PIP2. Localization studies using specific probes and immunostaining revealed that AU1 is predominantly located in nuclear speckles, whereas AU2 is distributed in different nuclear compartments, including the PNC, a compartment previously associated with active transcription and connected to cancer progression [58], where it colocalizes with PIP2 [12].

## 5. Functional Roles of Lipid−lncRNA Associations

Interactions between RNA and lipids exert diverse effects on essential biological processes, including cancer progression, autophagy, cellular compartment formation, and lipid homeostasis. LncRNAs have emerged as critical players in cancer progression, often functioning through pathways involving direct or indirect interactions with lipids and biomolecules. For example, LINK-A lncRNA promotes cancer progression and confers resistance to AKT inhibitors by hyperactivating AKT signaling. This occurs because PIP3, a key signaling lipid, binds to the pleckstrin homology (PH) domain of inactive AKT, inducing a conformational change that exposes its phosphorylation sites, Ser473 and Thr308, enabling its activation by upstream kinases such as PDK1 and the mTOR complex (Figure 2a) [59]. The lncRNA plays a critical role in this dynamic by interacting with AKT and PIP3 to form a complex that enhances this interaction. This activated form also phosphorylates GSK-3β, a protein related to processes such as apoptosis, reducing its kinase activity and leading to cell proliferation. The absence of this lncRNA could lead to a reduction in AKT phosphorylation, leading to the lack of activation necessary to phosphorylate other proteins (Figure 2b) [47].

A complex formation of RNA−lipid-protein is a mechanism also seen with the lncRNA SNHG9 that has a direct interaction with PA and the protein LATS1, promoting the phase separation of this protein, which affects the availability of LATS1 and inhibits the Hippo pathway, which is key to the regulation of cell proliferation [55]. In this pathway, the yes-associated protein (YAP) is phosphorylated by LATS1 and retained in the cytoplasm instead of translocating to the nucleus (Figure 3b). SNHG9 interaction with the PA of LATS1 plays a tumor-promoting role by affecting LATS1 phosphorylation by MOB1 and the subsequent phosphorylation of YAP. This interaction promotes the LLPS (Figure 3a) and inhibits LATS1 kinase activity, resulting in a non-phosphorylated YAP that is now able to translocate to the nucleus and associate with TEAD transcription factors [55].

HANR is an lncRNA that plays a role in the progression of several types of cancer, such as liver cancer [60,61], lung cancer [62], and colorectal cancer [63], and also intervenes in glucose metabolism in TNBC cells [64] by interacting with key regulatory molecules such as microRNAs and proteins such as hexokinase 2. HANR was recently found to colocalize in the nucleus with PIP2 (Figure 4a) [12,65] and also with proteins such as SON, MED1, and hnRNPI in the PNC of U2OS cells (Figure 4b). Where it may have a role as a structural component of the PNC and it is also suggested that it could possibly contribute to the phase separation of oncogenic genomic loci. The colocalization of AU1 in nuclear speckles and the colocalization of the AU2 motif mainly in the PNC compartment were also described, similar to HANR, where it is hypothesized that the AU2-PIP2 complex may play a role in transcription-dependent chromatin organization in the PNC [12].

Plasma membrane-associated RNA AL121772.1 (PMAR72) is an lncRNA found in the analysis of plasma membrane-associated RNAs and is thought to be involved in the formation of specific structures in the plasma membrane, as seen in an analysis of hybridization probes of PMAR72 that showed the formation of clusters on the plasma membrane [52]. Lipid membranes can affect RNA catalytic activity and serve as platforms for modulating RNA function. The effects of lipids on the activity of the R3C ligase ribozyme and its interactions with various RNA substrates were also analyzed. The presence of guanine and the formation of base-paired structures affected lipid binding. These findings suggest that lipid-sensitive RNAs could potentially be explored as tools for regulating RNA activity in a lipid-responsive manner and that these RNA-gel membrane interactions could be used for riboregulation [48]. To better understand the biological processes in which PI3P is implicated, such as autophagy, an aptamer PRA was developed that binds specifically to this phosphoinositide. In addition to being used as a possible imaging tool for the location of intracellular PI3P, the expression of PRA in HT1080 cells allowed us to observe the effects of this aptamer and has been characterized as an autophagy inhibitor for the specific interaction with this phosphoinositide and control its participation and also as a possible tool to develop drugs to control autophagy [57].

LIPTER, a newly identified long non-coding RNA, plays a crucial role in maintaining cardiac lipid metabolism homeostasis. By interacting with the mobility protein MYH10 and the phospholipids PA and PI4P present on the surface of LD, LIPTER regulates lipid synthesis and metabolism, protecting the heart from lipid overload and thereby preventing cardiac dysfunction by acting as a scaffold between LD and the transport protein and being delivered to processing sites such as mitochondria (Figure 5a). In case of the lack or silencing of LIPTER, it may result in the accumulation of lipid droplets and potential complications associated with lipotoxicity in the cell (Figure 5b) [13].

Fundamentally, the interactions between lipids and long non-coding RNAs play significant roles in diverse cellular processes, including cancer, autophagy, and lipid metabolism. While some of these lncRNAs already have a defined nucleotide sequence responsible for the interaction, as shown in Table 1, further research is needed to fully understand the dynamics of these interactions.

The potential applications of lipid–RNA interactions are broad, with lipid-binding RNAs being explored not only in the analysis of extracellular vesicle-associated RNAs but also as biosensors for monitoring viral RNAs in biofluids and as carriers for delivering therapeutic RNAs in gene therapy approaches [51]. Recent technological advances, such as NANOSPRESSO, demonstrate the growing feasibility of locally producing nucleic acid-based nanomedicines with high precision. These tools open new possibilities for the therapeutic exploration of lncRNA−lipid interactions [66]. Understanding the mechanisms by which certain lncRNAs interact with lipids not only expands our knowledge of cellular regulation but also opens up promising therapeutic avenues. For instance, lncRNAs such as LINK-A have been shown to modulate pathways involved in cancer progression and drug resistance [47], positioning them as potential therapeutic targets in oncology. Similarly, LIPTER plays a critical role in preventing lipotoxicity [13]; thus, modulating lncRNA activity presents a potential avenue for treating metabolic disorders. These findings underscore the need for further investigation into lipid–lncRNA interactions, which may yield innovative strategies for targeted therapies and precision medicine, particularly in diseases characterized by the convergence of lipid dysregulation and RNA-based mechanisms.

## 6. Conclusions

The evidence reviewed here indicates a growing recognition of the importance of lipid−lncRNA interactions and their impact on various biological processes. These interactions are emerging as novel regulatory mechanisms that can be exploited for potential therapeutic applications. The physicochemical properties of lipids provide an environment for a physical location that can dynamically alter the structural arrangement and function of lncRNAs. Although many lncRNAs have been found to interact with lipids, the molecular mechanisms underlying these interactions remain unclear. Future research should focus on elucidating the specific mechanisms by which lncRNAs and lipids interact and regulate cellular processes, as well as how chemical modifications involving phosphorylation, methylation, and acetylation, which affect the physicochemical properties of the modified molecules, may affect their interaction with lipids and lncRNA complex formation. With the increasing understanding of the functional roles of lipid−lncRNA interactions, there is a growing potential to develop novel therapeutic strategies targeting these interactions for the treatment of various diseases. Liquid−liquid phase separation involving lncRNAs and lipids is an emerging area that warrants further investigation, and how the polymeric structure of RNAs may provide an additional scaffold for structural organization that allows for more complex regulation and organization.

The visualization and quantification of different lipid species within the cell nucleus remain highly limited. While lipids are known to participate in a wide variety of membrane-less structures within the nucleus, their study faces significant constraints. Unlike RNA dynamics, which can be monitored in real time using fluorescent aptamers, lipid visualization remains challenging due to the lack of comparable tools and methodological limitations.

Novel RNA–lipid interaction profiling methods, like LIPRNA-seq, significantly enhance our understanding of the variety of RNA species that can specifically interact with phospholipids, many of which were previously unassociated with such activities. These findings indicate that RNA–lipid interactions may be more widespread and functionally relevant than previously thought. Although the exact functions of these RNAs in regulation, structural organization, or spatial compartmentalization are not yet completely understood, their identification provides new opportunities to explore the interplay between RNA and lipid-based cellular signaling.

The application of techniques designed to identify lncRNA−lipid interactions may reveal previously unrecognized molecular functions of lncRNAs. Many lncRNAs known to participate in processes such as lipid metabolism may also be involved in lipid regulation through direct or indirect interactions that have yet to be elucidated. This broader perspective is supported by discoveries such as LINK-A, which emerged as a novel participant within pathways previously considered well-characterized, revealing additional regulatory roles and molecular partners. Given the regulatory roles of lncRNAs in lipid metabolism and the ability of some to associate with specific phosphoinositides, these molecules could be exploited as therapeutic agents or delivery modulators. LncRNAs, such as LIPTER and LINK-A, have illustrated how lipid-associated mechanisms can uncover unexpected layers of regulation, expanding our understanding of diseases, such as cancer and metabolic disorders. Integrating lncRNA−lipid knowledge into nanomedicine design could offer novel avenues for precision treatments.

## Figures and Tables

**Figure 1 cells-14-01193-f001:**
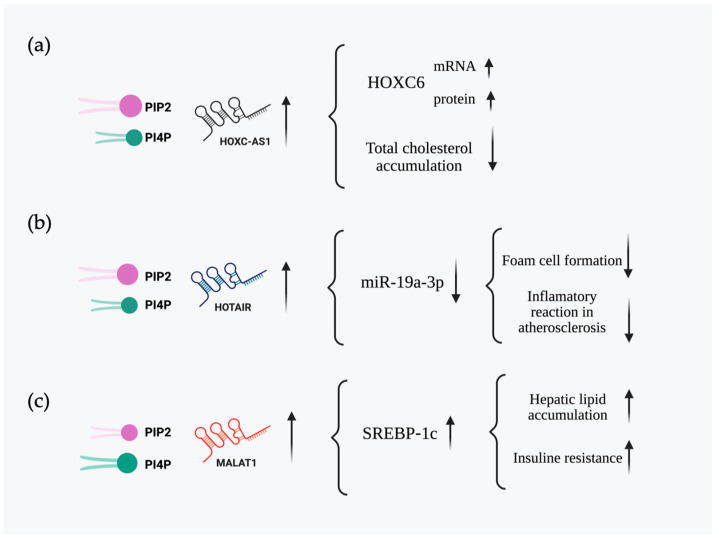
LncRNAs involved in lipid metabolism identified in a database of RNAs interacting with PIP2 and PI4P. On the left side of each panel, the specific phosphoinositide with which each lncRNA interacts is indicated, as well as their relative binding affinity, represented by the larger lipid icon corresponding to the stronger interaction. The lncRNAs shown with their corresponding influence in the lipid metabolism are: (**a**) HOXC-AS1, (**b**) HOTAIR, and (**c**) MALAT1.

**Figure 2 cells-14-01193-f002:**
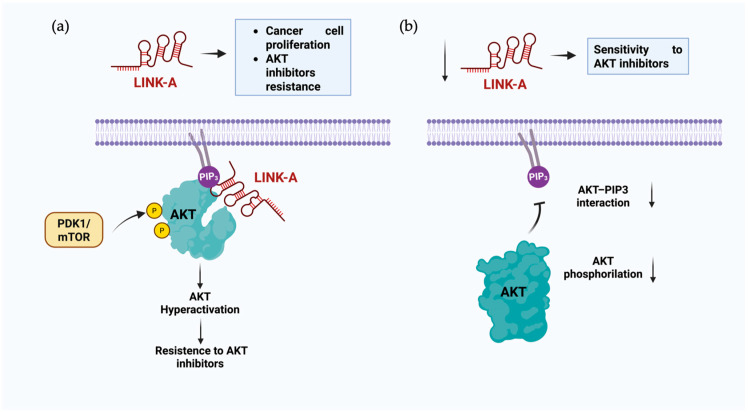
Proposed model of potential biological functions of LINK-A and PIP3 interaction. (**a**) LINK-A interaction within the complex formed by AKT enhances the interaction with PIP3 and possibly allows a conformational change in AKT, improving its phosphorylation and leading to AKT hyperactivation. (**b**) The absence of LINK-A could diminish the interaction between AKT and PIP3 and phosphorylation by other proteins.

**Figure 3 cells-14-01193-f003:**
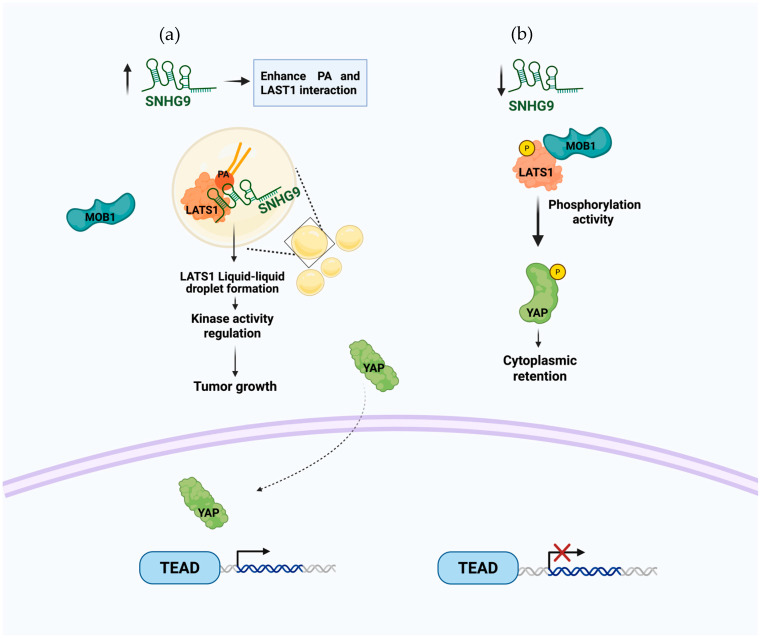
Proposed model of the potential biological functions resulting from the interaction between SNHG9 and PA. (**a**) The interaction of SNHG9 with PA and the LATS1 protein leads to the formation of liquid droplets. This diminishes the activation and availability of LATS1 to phosphorylate YAP, allowing its translocation to the nucleus and interaction with TEAD transcription factor. (**b**) The absence of SNHG9 enables the phosphorylation of LATS1 by other proteins and also allows it to phosphorylate the YAP protein, promoting its cytoplasmic retention.

**Figure 4 cells-14-01193-f004:**
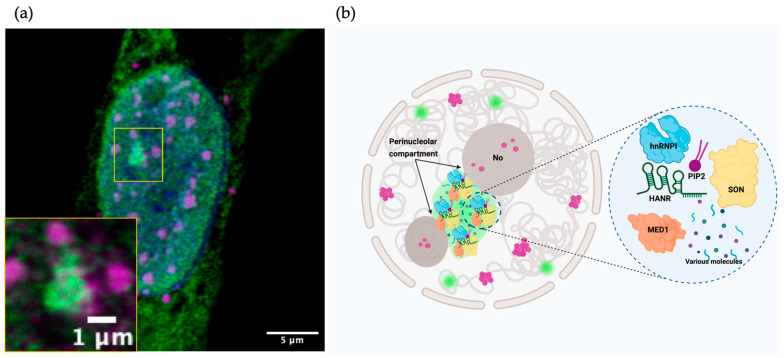
(**a**) FISH assay of HANR and PIP2 immunolocalization in U2OS cells (modified from Bayona-Hernandez, 2024) and (**b**) a model of HANR accumulation within perinucleolar compartments, where it colocalizes with PIP2 as well as with several key proteins, including MED1, SON, and hnRNPI.

**Figure 5 cells-14-01193-f005:**
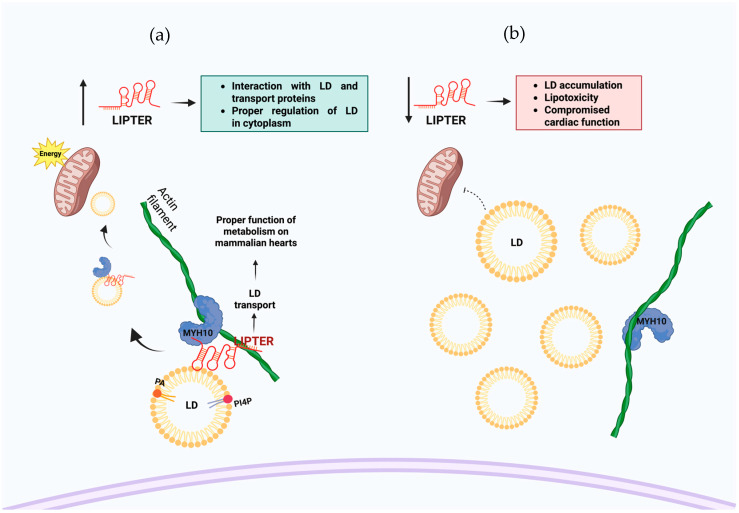
Proposed model of potential biological functions of LIPTER and lipids on the surface of lipid droplets involving PI4P and PA interactions. (**a**) LIPTER can interact with phospholipids on the LD surface and with the MYH10 protein, which transports the LDs along actin fibers toward the mitochondria. (**b**) The absence of LIPTER may result in the accumulation of lipid droplets and potential complications associated with lipotoxicity.

**Table 1 cells-14-01193-t001:** Summary of RNA−lipid interactions. The table lists various RNA molecules along with their corresponding lipid interactors, potential functions, and the specific sequences responsible for these interactions.

RNA Name	Lipid Interaction	Function	Sequence or Motif Region of Interaction	Reference
HANR	Phosphatidylinositol 4,5-bisphosphate	Possible contribution to the formation and stabilization of phase-separated structures such as the perinucleolar compartment (PNC)	-	[12]
AU1		-	AUGUAAAAAAAAAAUUAAAGAAAAAAAAAAAAAUAAAAA	
AU2		-	UUUUUCAAGAGUGAUAAAAAGUUUUUGGCCC	
LIPTER (LINC00881)	Phosphatidic acid and phosphatidylinositol 4-phosphate	Improves the LD transport by MYH10, PA, and PI4P binding	342–791 nt (Exon 3)	[13]
LINK-A	Phosphatidylinositol-3,4,5-trisphosphate	Hyperactivation of AKT confers resistance to AKT inhibitors	1081–1140 nt. Loop sequence: 5′-CAGGGUAGACUCGCUCUG-3′ (1100–1117 nt)	[47]
	Phosphatidylcholine (PC)	-	241–300 nt	
Random RNA sequences	Gel phase [dipalmitoylphosphatidylcholine (DPPC) and other phosphatidylcholine lipids	As a tool for the development of synthetic riboswitches and lipid biosensors.	High Guanine content sequence	[48]
PMAR72	Sphingomyelin (SM), cholesterol, and phosphatidylinositol-3,4,5-trisphosphate	Suspected to be involved in the specific cell membrane structures	-	[52]
SNHG9	Phosphatidic acid (PA)	Facilitates the formation of liquid droplets of Large Tumor Suppressor Kinase 1 (LATS1)	204–231 nt	[55]
PI3P RNA aptamer (PRA)	phosphatidylinositol 3-phosphate	Imaging tool for PI3P localization in the cell and as an inhibitor in the autophagy process.	CAUUCGCAGGGAGUUAUAGCAGAGGAAUACAAAUAGGGGG (40 nt)	[57]

## Data Availability

Not applicable.

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
