# Peer review of "Lipid−lncRNA Crossroads: An Overview of Interactions Between Lipids and lncRNA"

_cells, 2025, doi:10.3390/cells14151193_

Round 1
Reviewer 1 Report
Comments and Suggestions for Authors
The review by Bayona-Hernandez addresses lncRNAs and their interactions with lipids and involvements in LLPS.
The review has an attractive and timely topic, is well structured by its headings and coverse many relevant examples of lncRNAs. However, in its current version, the text reads in many passages as a summary of findings without comparing, contrasting or evaluating the evidences of the cited publications.
I have the following points:
1. The manuscript lacks criticial analysis and discussion. Throughout the manuscript, but especially in the lncRNA parts, the examples are presented as boring summary of papers. It would be helpful if the authors can also evaluate, compare and contrast different lncRNAs-lipid interaction mechanisms.
2. A discussion about limitations of the current methods would be helpful. What do we really need to push the field forward?
3. For many chapters, the authors start right away with examples. Introductory sentences would help to introduce the topic of the chapter. Additionally, at the end of each chapter, a summary or concluding sentences of what the authors have provided is missing. What do we learn from each chapter?
4. For chapter3 about LLPS, i miss an explanation about A/B compartments and lncRNAs involved in it. I think this chapter needs more detail. How do lipids contribute to lncRNA/protein phase separation? Are intrinsically disordered domains important for interaction with lncRNAs? What is their role in LLPS?
5. Introduction: Reference 1 seems not appropriate as this review is originally from plant sciences. Please replace. Additionally, the authors forgot to mention that lncRNAs are also physiologically important. Please include for this statement a publication.
6. The role of HANR in phase separation seems to be too speculative resulting from overinterpretation. Please revise or weaken the statement and figure.
7. The conclusion section is weak and reads as if the introduction is continued. It needs future directions, unanswered questions and ideas for new techniques. For example, lncRNAs are often cell type-specific and lack high levels of conservation. Does that mean that lipid interactions are lost or compensated or also conserved?
8. Are lncRNA-lipid interactions and the formation of LLPS important also for other diseases than cancer? I understand that most of the work was done in cancer, but it would be great if the authors could find more examples for other diseases.
Author Response
Open Review
Reviewer 1
(x) I would not like to sign my review report
( ) I would like to sign my review report
Quality of English Language
( ) The English could be improved to more clearly express the research.
(x) The English is fine and does not require any improvement.
Comments and Suggestions for Authors
The review by Bayona-Hernandez addresses lncRNAs and their interactions with lipids and involvements in LLPS.
The review has an attractive and timely topic, is well structured by its headings and coverse many relevant examples of lncRNAs. However, in its current version, the text reads in many passages as a summary of findings without comparing, contrasting or evaluating the evidences of the cited publications.
I have the following points:
1. The manuscript lacks criticial analysis and discussion. Throughout the manuscript, but especially in the lncRNA parts, the examples are presented as boring summary of papers. It would be helpful if the authors can also evaluate, compare and contrast different lncRNAs-lipid interaction mechanisms.
We appreciate the reviewer's insightful comments and have incorporated additional text and information throughout the manuscript as suggested. In particular, we have expanded the discussion to include further details on lncRNA-lipid interaction mechanisms.
- A discussion about limitations of the current methods would be helpful. What do we really need to push the field forward?
The observation and detection of different lipid species within the cell nucleus remain highly limited. While lipids are known to participate in a wide variety of membraneless structures within the nucleus, their study faces significant constraints. Unlike RNA dynamics, which can be monitored in real time using fluorescent aptamers, lipid visualization remains challenging due to the lack of comparable tools and methodological limitations. Into the conclusion section.
- For many chapters, the authors start right away with examples. Introductory sentences would help to introduce the topic of the chapter. Additionally, at the end of each chapter, a summary or concluding sentences of what the authors have provided is missing. What do we learn from each chapter?
We have added a short sentence to improve the coherence of the chapters.
- For chapter3 about LLPS, i miss an explanation about A/B compartments and lncRNAs involved in it. I think this chapter needs more detail. How do lipids contribute to lncRNA/protein phase separation? Are intrinsically disordered domains important for interaction with lncRNAs? What is their role in LLPS?
We added more detail about intrinsically disordered proteins into phase separation in section 3. Interaction platforms for RNA-lipid-protein in liquid-liquid interactions.
- Introduction: Reference 1 seems not appropriate as this review is originally from plant sciences. Please replace. Additionally, the authors forgot to mention that lncRNAs are also physiologically important. Please include for this statement a publication.
We changed the reference for a more adequate to the topic of the paper.
- The role of HANR in phase separation seems to be too speculative resulting from overinterpretation. Please revise or weaken the statement and figure.
We took that into consideration and made significant modification to the image to match the statement of the colocalization of HANR with lipids and other proteins of interest.
- The conclusion section is weak and reads as if the introduction is continued. It needs future directions, unanswered questions and ideas for new techniques. For example, lncRNAs are often cell type-specific and lack high levels of conservation. Does that mean that lipid interactions are lost or compensated or also conserved?
Following the suggestion, we've significantly expanded the conclusions section to explore in more detail the potential applications and significance of the ncRNA-phospholipid interactions discussed in the review.
- Are lncRNA-lipid interactions and the formation of LLPS important also for other diseases than cancer? I understand that most of the work was done in cancer, but it would be great if the authors could find more examples for other diseases.
Besides Cancer related there is also the example of the lncRNA LIPTER which is related to the lipid droplet transport to avoid lipotoxicity in the cells. Also, not a lncRNA, but the aptamer PRA showed that could be related to autophagy.
Submission Date
09 July 2025
Date of this review
17 Jul 2025 17:04:26
Reviewer 2 Report
Comments and Suggestions for Authors
The manuscript by Bayona-Hernandez and collaborators concerns the role of interactions of lncRNA with lipids. The authors provide a rather general overview of the topic while focusing on selected mechanistic aspects and functional roles of such interactions. The subject is of potentially of broad interest and also nicely fits into the trends of recent years. It is worth stressing that the authors are experts in the field and performed some pioneering studies themselves. Surely, the content is original and thus far underrepresented in other review papers. On the other hand, in the current shape the whole story is presented in a rather chaotic way, which makes it difficult for the reader to draw constructive conclusions. Therefore, before publishing the work the authors should reconsider the general layout and more carefully arrange the content. There are also a few more specific issues listed below.
- The text that deals with lipids (l. 96 – 118) should be more thoroughly elaborated and most preferentially presented as a separate section (e.g. chapter 2).
- All the lipids mentioned in the chapters 3-5 should be properly introduced. Why not outlining their general physiological role in a few sentences?
- The discussion about RNA sequences and structural motifs responsible for lipid binding is rather insufficient in its current form. For example, formation of high-molecular-weight complexes (e.g. DOI: 10.1023/A:1016063414162) and presence of divalent ions which change behaviour of zwitterionic lipids and thus interactions with nucleic acids (e.g. DOIs: 10.1021/la0531796; 1021/acs.jpcb.5b01256) are important issues. RNAs can also influence membrane permeability (e.g. DOI: 10.1073/pnas.96.19.10649). Moreover, it is not only gel phase membranes which exhibit increased RNA binding, since higher affinity of RNAs towards ordered lipid membranes compared to liquid disordered membranes was also observed (e.g. DOI: 10.1038/s41598-025-91093-x).
- For sake of clarity specific molecules (e.g. PCSK9) and cells (e.g. THP-1 macrophages) should be properly introduced in the text.
- Nomenclature of lipids is to a large extent inconsistent. For example, within the text one may find “PI(4)P” “PI4P” or “PIP4”, all defining the same lipid. Similarly, “PI(4,5)P2” is also inappropriately defined as “PI(4,5P)2” or ambiguously “PIP2”. Regarding this issue, the authors should follow recent international standards (i.e. DOI:10.1194/jlr.S120001025)
- The text from l.239-248 requires clarification in light of the content of the figure 1.
- In general, the references to the figures in the main text are often wrong (e.g. l. 248, 279, 307, etc.).
- There are also lots of typos which require correction (e.g. missing or redundant spaces in l. 78, 82, 92, 95, 160,192, 241, 276, periods l.147, etc.). References in the table 1 should be provided in the same formatting as in the main text.
Author Response
The manuscript by Bayona-Hernandez and collaborators concerns the role of interactions of lncRNA with lipids. The authors provide a rather general overview of the topic while focusing on selected mechanistic aspects and functional roles of such interactions. The subject is of potentially of broad interest and also nicely fits into the trends of recent years. It is worth stressing that the authors are experts in the field and performed some pioneering studies themselves. Surely, the content is original and thus far underrepresented in other review papers. On the other hand, in the current shape the whole story is presented in a rather chaotic way, which makes it difficult for the reader to draw constructive conclusions. Therefore, before publishing the work the authors should reconsider the general layout and more carefully arrange the content. There are also a few more specific issues listed below.
- The text that deals with lipids (l. 96 – 118) should be more thoroughly elaborated and most preferentially presented as a separate section (e.g. chapter 2).
It was moved to the following section.
- All the lipids mentioned in the chapters 3-5 should be properly introduced. Why not outlining their general physiological role in a few sentences?
We addressed this and the information was expanded in the 3 section.
- The discussion about RNA sequences and structural motifs responsible for lipid binding is rather insufficient in its current form. For example, formation of high-molecular-weight complexes (e.g. DOI: 10.1023/A:1016063414162) and presence of divalent ions which change behaviour of zwitterionic lipids and thus interactions with nucleic acids (e.g. DOIs: 10.1021/la0531796; 1021/acs.jpcb.5b01256) are important issues. RNAs can also influence membrane permeability (e.g. DOI: 10.1073/pnas.96.19.10649). Moreover, it is not only gel phase membranes which exhibit increased RNA binding, since higher affinity of RNAs towards ordered lipid membranes compared to liquid disordered membranes was also observed (e.g. DOI: 10.1038/s41598-025-91093-x).
We agree and addend more detail information. However one of the previous citations here was with DNA therefore we added a recently accepted article that is specific to this type of interactions. Nevertheless much of the information that is published focus on RNA interacting with lipids in membranes. Here however, we like to highlight those lipids that are in membraneless structures such as nucleoli, Cajal bodies etc. Also along the text we include more of the suggested papers to improve the discussion and understanding of the topic, mainly in section 4.
- For sake of clarity specific molecules (e.g. PCSK9) and cells (e.g. THP-1 macrophages) should be properly introduced in the text.
We have speficy cell type, like Hella cells with HANR and PIP2. On the figure 5 legend
- Nomenclature of lipids is to a large extent inconsistent. For example, within the text one may find “PI(4)P” “PI4P” or “PIP4”, all defining the same lipid. Similarly, “PI(4,5)P2” is also inappropriately defined as “PI(4,5P)2” or ambiguously “PIP2”. Regarding this issue, the authors should follow recent international standards (i.e. DOI:10.1194/jlr.S120001025)
The manuscript has been homogenize into current nomenclature used in the literature today with the exception of the introduction full name of the lipids at the beginning of each lipid used in this review.
- The text from l.239-248 requires clarification in light of the content of the figure 1. We agree and have clarified the text
- In general, the references to the figures in the main text are often wrong (e.g. l. 248, 279, 307, etc.).
It has been corrected
- There are also lots of typos which require correction (e.g. missing or redundant spaces in l. 78, 82, 92, 95, 160,192, 241, 276, periods l.147, etc.). References in the table 1 should be provided in the same formatting as in the main text. We have fixed all the typos in the text
Reviewer 3 Report
Comments and Suggestions for Authors
This review addresses a frontline argument of research: the interaction of lipid and longRNAs. English is fluent and paer well organized but a tendency to “listing” is present. It is suggested to try to insert the summaries of relevant articles in a more fluent argumentation.
However, my main perplexity is that the authors keep on mixing two interesting but completely distinct “interaction” concepts:
- Physical interaction of lncRNA and lipids, a molecular mechanisms controlling several cellular biological aspects. This is described with molecular details and represent the core of the article
- The functional control that lncRNAs exert on lipid metabolism by directing transcription and translation of specific RNAs. This is summarized with some examples but no specific mechanism, my personal idea is to remove this part and maybe deal with it in a future work.
Minor points:
Figure 3 contains a picture, please credit author and methods
Table 1 format (function column) is unsatisfactory and hard to read
The conclusion/discussion section could be expanded and made more speculative regarding future developments and general conclusions.
Author Response
- Physical interaction of lncRNA and lipids, a molecular mechanisms controlling several cellular biological aspects. This is described with molecular details and represent the core of the article
- The functional control that lncRNAs exert on lipid metabolism by directing transcription and translation of specific RNAs. This is summarized with some examples but no specific mechanism, my personal idea is to remove this part and maybe deal with it in a future work.
We appreciate the reviewer’s insightful comment and understand the concern regarding the coexistence of two distinct interaction concepts within the manuscript. The section 2 discusses several lncRNAs present in a database obtained from a previous study, where these lncRNAs were found to interact with PIP2, which we have addressed recently in the text. While the study focused on the relationship with other lipids and did not directly demonstrate physical interactions, it was interesting to note that they were related in some way to other lipid metabolism processes or cells directly related as adipocytes. This information was meant to provide biological context and highlight that the lncRNA–lipid relationship is not limited to physical binding. Although this approach might have introduced confusion and detracted from the core focus on physical interactions, specifying the intention of the section clarifies that it serves as background information regarding their relationship, rather than direct interaction itself.
Minor points:
Figure 3 contains a picture, please credit author and methods
We have Addressed in the figure
Table 1 format (function column) is unsatisfactory and hard to read
The table format is given by MDPI
The conclusion/discussion section could be expanded and made more speculative regarding future developments and general conclusions.
We expanded the conclusion section, we thank you for the input.
Reviewer 4 Report
Comments and Suggestions for Authors
The manuscript by Bayona-Hernandez and colleagues deal with the interaction between lncRNA and lipids. The topic is of general interest and many papers are coming out to deal with this aspect of lncRNA world.
Major concern:
- Add a paragraph and a figure that recapitulate the lipid metabolism playing particular attention on the lipids that are mentioned in the text. In this paragraph please transfer the lanes 96-118 that descried general roles of lipids.
- The separation between lncRNA regulation of lipid metabolism and lncRNA interaction with lipids should be more clear.
- Add a paragraph on LncRNA and extracellular vescicles (for a reference see doi: 10.3390/cancers17132186)
- The addition of a paragraph on the therapeutical potential of LncRNA-lipid interaction will be also very interesting to broaden the readership (see for example the doi: 10.3389/fsci.2025.1458636)
Minor point:
- The paper is in general well written even if I strongly recommend to carefully check sentence structure along all the manuscript (i.e. pag7 lanes 239-245).
- Figure 3 is not cited in the text, while figure 4 is mixed up with figure 3 (lane 307)
Author Response
The manuscript by Bayona-Hernandez and colleagues deal with the interaction between lncRNA and lipids. The topic is of general interest and many papers are coming out to deal with this aspect of lncRNA world.
Major concern:
- Add a paragraph and a figure that recapitulate the lipid metabolism playing particular attention on the lipids that are mentioned in the text. In this paragraph please transfer the lanes 96-118 that descried general roles of lipids.
We have expanded this in the third section
- The separation between lncRNA regulation of lipid metabolism and lncRNA interaction with lipids should be more clear.
We have added additional information to improve this point.
- Add a paragraph on LncRNA and extracellular vescicles (for a reference see doi: 10.3390/cancers17132186)
Extracellular vesicles are lipid-bilayer-delimited particles released by cells, known to transport bioactive molecules such as proteins, DNA, lipids, and RNA, including long non-coding RNA (lncRNA) and circular RNA (circRNA). They facilitate intercellular communication and play a crucial role in both physiological effects and certain pathological processes. Specific lncRNAs can be selectively packaged into EVs, where they can modulate gene expression in recipient cells at multiple levels, thereby influencing processes such as cancer progression [1]. RNA structural motifs have been found to modulate the interaction with the plasma membrane, suggesting the importance of these motifs for the loading of lncRNAs into extracellular vesicles and RNA-based lipid biosensors. Analysis on Y RNA has been described to contain the UCCCU RAFT motif, which promotes lipid raft association, and the GGAG EXO motif, known to facilitate miRNA sorting into exosomes. Furthermore, the presence of multiple small apical loops within the RNA structure was associated with an enhanced affinity for RAFT liposomes. Also was found that in viral RNA fragments with long double helix at the apical loop increases the affinity of viral RNA to lipid rafts [2]
- The addition of a paragraph on the therapeutical potential of LncRNA-lipid interaction will be also very interesting to broaden the readership (see for example the doi: 10.3389/fsci.2025.1458636)
The possible applications as the types of different lipid RNA binding are wide, into the analysis of RNA for extracellular vesicles was explore the use as biosensors to monitoring viral RNAs in biofluids, as well as the delivery of therapeutic RNAs for the use in gene therapy
Minor point:
- The paper is in general well written even if I strongly recommend to carefully check sentence structure along all the manuscript (i.e. pag7 lanes 239-245). We thank the reviewer and check the structure to improve the manuscript
- Figure 3 is not cited in the text, while figure 4 is mixed up with figure 3 (lane 307)
We have fixed the problem with the mix figs
Round 2
Reviewer 1 Report
Comments and Suggestions for Authors
I have no further points.
Reviewer 2 Report
Comments and Suggestions for Authors
I would like to thank the authors for addressing all the issues raised by reviewers. The manuscript is now ready for publication.
Reviewer 3 Report
Comments and Suggestions for Authors
Author's responses fully address my concerns. I do not noted further critical points